# Real-Time Energy Data Acquisition, Anomaly Detection, and Monitoring System: Implementation of a Secured, Robust, and Integrated Global IIoT Infrastructure with Edge and Cloud AI

**DOI:** 10.3390/s22228980

**Published:** 2022-11-20

**Authors:** Raihan Bin Mofidul, Md. Morshed Alam, Md. Habibur Rahman, Yeong Min Jang

**Affiliations:** Department of Electronic Engineering, Kookmin University, Seoul 02707, Republic of Korea

**Keywords:** industrial internet of things, message queuing telemetry transport secured, heterogeneous data extraction, global interconnection, edge and cloud AI, real-time monitoring, anomaly detection

## Abstract

The industrial internet of things (IIoT), a leading technology to digitize industrial sectors and applications, requires the integration of edge and cloud computing, cyber security, and artificial intelligence to enhance its efficiency, reliability, and sustainability. However, the collection of heterogeneous data from individual sensors as well as monitoring and managing large databases with sufficient security has become a concerning issue for the IIoT framework. The development of a smart and integrated IIoT infrastructure can be a possible solution that can efficiently handle the aforementioned issues. This paper proposes an AI-integrated, secured IIoT infrastructure incorporating heterogeneous data collection and storing capability, global inter-communication, and a real-time anomaly detection model. To this end, smart data acquisition devices are designed and developed through which energy data are transferred to the edge IIoT servers. Hash encoding credentials and transport layer security protocol are applied to the servers. Furthermore, these servers can exchange data through a secured message queuing telemetry transport protocol. Edge and cloud databases are exploited to handle big data. For detecting the anomalies of individual electrical appliances in real-time, an algorithm based on a group of isolation forest models is developed and implemented on edge and cloud servers as well. In addition, remote-accessible online dashboards are implemented, enabling users to monitor the system. Overall, this study covers hardware design; the development of open-source IIoT servers and databases; the implementation of an interconnected global networking system; the deployment of edge and cloud artificial intelligence; and the development of real-time monitoring dashboards. Necessary performance results are measured, and they demonstrate elaborately investigating the feasibility of the proposed IIoT framework at the end.

## 1. Introduction

The industrial internet of things (IIoT) is a system of interconnected devices used in industrial settings to monitor and control machinery, production lines, and human labor in real time to boost efficiency. The notion of “Industry 4.0” refers to a subset of the IIoT that places an emphasis on worker protection and increased output [1]. Nowadays, the IIoT infrastructure is driven by the internet of things (IoT), cloud and edge computing, cyber security, AI and machine learning, and digital twin [2]. In order to decrease failures and save time and investment, companies are considering AI-powered visual insights to replace manual inspection business models. Such as in [3], a classification model between microseismic and blasts events using the convolutional neural network (CNN) was proposed to analyze the mechanical parameters contained in microseismic events for providing accurate information of rockmass. Manufacturers can use machine learning algorithms to detect problems as soon as possible [4]. On the other hand, “Industry 5.0” refers to a future workplace environment in which humans and smart robots coexist. Industry 5.0 aims to combine cognitive computing capabilities with human intellect and resourcefulness in collaborative operations as robots in the workplace become more intelligent and interconnected [5]. AI, big data, supply chain, digital transformation, machine learning, and the IoT have all been identified as some of the most popular and widely used enablers for Industry 5.0 [6]. Moreover, the IoT-enabled industries have a big impact on the environment since they use scarce resources and lots of energy during production, usage, and recycling. In response, the area of research known as the green IoT has emerged to reduce this carbon effect [7]. The “industrial revolution” is propelled by increased connectivity, openness of data, decentralized and automated decision making, and technological support. Industries may now collect and analyze data in real time through IoT systems for monitoring, exchanging data, and evaluating the state of the environment. When it comes to the IIoT, speed and efficiency are paramount. Large-scale deployments are required for complex systems. Therefore, it is essential that sensors maintain their performance over time while keeping costs reasonable. If the information from these sensors is utilized to make important choices, then latency is a measure of performance. As a popular protocol for the IoT, the message queuing telemetry transport (MQTT) is highly regarded. It is flawless because of its small code size, seamless integration, and outstanding performance [8]. In addition, an essential feature of the IIoT for cyber–physical systems is the capacity for near real-time data streaming, which is necessary for the seamless integration of the physical and digital worlds. The manufacturer may obtain valuable insights from the acquired data. It is also possible to utilize the data to spot subtle problems with the manufacturing facility’s infrastructure. Furthermore, the data may be used for improvement and prediction, giving the data from the IoT devices real value. Eighty-four percent of businesses surveyed for their big data and cloud strategy cited the need for a unified platform to facilitate the transfer of information to the cloud as a top priority [9]. Furthermore, the manufacturing industry must modify its practices in response to reducing manpower, economic convenience, and ecological norms. Management of production needs adaptable decision-making procedures and the ability to self-configure. Data collected in real time from the factory floor may help guide strategy. Through real-time monitoring, any advanced system in the IIoT may make choices and delegate authority to various stakeholders in an organization so that they can act on data in real time [10].

However, there are grave concerns relating to energy savings, real-time performance, cohabitation, compatibility, security, and privacy in the adoption of the “Industry 4.0” level IIoT infrastructure [11]. In Ref. [12], the healthcare industrial IoT (HealthIIoT) was proposed to monitor, track, and store patients’ healthcare information for continuous care, with data watermarked before being sent to the cloud for secure, safe, and high-quality health monitoring. However, they did not utilize any AI algorithms or features. According to [9], service-oriented architecture (SOA) was introduced to handle the heterogeneous data of IoT and IIoT devices. However, they were unable to provide enough details and an appropriate solution for edge IoT sensors that communicate securely with a cloud server. Reference [10] proposed methods for facilitating the digital transformation of a manufacturing line and tying such methods into the concept of the digital twin. Methods for implementing online monitoring using both traditional and IIoT sensors and collecting the resulting data were discussed. However, this article did not go into sufficient detail on the edge computing devices and the interconnection of the vast IIoT networking architecture. Identical articles, such as [13,14,15,16,17,18,19], proposed a three-terminal collaborative platform (TTCP), integration of AI and IIoT technologies, transparency relying upon statistical theory (TRUST), deep learning (DL), and AI-enabled software-defined IIoT network (AI-SDIN), to implement “Industry 4.0” and “Industry 5.0” facilities. Nevertheless, each of these approaches brings its own unique perspective, ignoring the global interconnected IIoT networking system. A LoRaWAN-based local IIoT infrastructure was introduced in [20] while the proposed system covers the global IIoT framework. In addition, the authors implemented a state-of-the-art open-source P2P energy trading platform in [21] that makes use of IoT and blockchain technology. It was unexpected to discover that they declared Node-RED as their MQTT broker, as Node-RED can only act as a MQTT client while making a connection with a MQTT broker service such as the Mosquitto MQTT broker according to Refs. [22,23]. Furthermore, their proposed system is neither https- nor MQTTS-enabled, and customers would have to pay for a limited number of infrastructure components, such as private blockchain service, to use it. On top of that, they have not integrated AI into their system. Similar articles, such as [24,25,26,27,28,29], introduced interesting technologies, such as augmented password-only authentication and key exchange (AugPAKE), attribute-based encryption (ABE), oblivious transfer (OT), generic MQTT protocol with Mosquitto broker, and so on. Each of these publications is unaware of the integrated global IIoT systems and open-source, such as openssl [30], based remarkable encryption protocols, such as utilizing self-certified certificates in TLS and SSL cryptographic protocols, which provide an extremely secure and incredibly fast communication system in an integrated IIoT infrastructure. Moreover, a simulation-based smart controller device was introduced in [31] for classifying the contracted load through a data-acquisition approach, whereas the proposed SDAD is integrated and implemented on a real system. The authors in [32] developed machine-learning-based abnormal voltage regulation detection in PV systems, where the proposed architecture is focused on anomaly data detection in every electrical appliance. For continuous energy flow monitoring purposes [33], the offered technique develops an AI integrated real-time monitoring system through the IIoT framework.

In this article, we implemented a globally distributed, secure, resilient, and integrated IIoT infrastructure for real-time energy data acquisition, management, monitoring, and anomaly detection. Edge and cloud AI were also integrated on the basis of “Industry 4.0” and “Industry 5.0” applications. Several algorithms, flow-charts, as well as customized devices such as SDAD were exposed. Multiple edge servers, a global MQTTS broker, and an integrated cloud server were developed. Open-source-based software such as Node-RED, Mosquitto, openssl, Visual Studio Code, etc., were utilized. In summary, the primary contribution of our research comprises the following:Design and development of smart data acquisition devices, which are used to measure the power consumption of home appliances, focused on keeping them compact, sturdy, and economical.Afterward, HTTPS-enabled edge servers utilizing Node-RED are built for acquiring data from SDADs and inserting these data into databases.Implementation of a TLS-enabled global MQTTS broker leveraging open-source software “Mosquitto” for sharing information between edge servers and cloud/centralized servers.Construction of SQL databases through “PostgreSQL” in order to handle heterogeneous big data.Incorporating edge and cloud AI into the system to identify outliers in the sensor readings.Finally, individual and centralized dashboards were implemented for real-time monitoring of the system.

On the basis of the above contributions, it is clear that our suggested system is highly advantageous in the IIoT system due to its simple architecture, secured and swift connectivity, processing capabilities of heterogeneous massive data, integration with AI, and real-time monitoring dashboards (that anyone with the proper credentials can access at any time, from any location). In addition, open-source software is used in every aspect of the proposed system, resulting in cost savings. The outline of the paper looks like the following: The proposed methodology is described in Section 2. Implementations of software and hardware are demonstrated in Section 3. Section 4 induces system evolution and experimental outcomes. In Section 5, a brief discussion and the future direction of this study are revealed.

## 2. Methodology

### 2.1. System Overview

Our developed system is made up of three fundamental parts: (a) smart data acquisition devices to obtain values from the sensor nodes, (b) edge IIoT systems to obtain different types of data of individual houses and run AI models to detect anomalies, and (c) a centralized IIoT system to analyze heterogeneous data of all houses, run AI models, and exchange necessary data with the edge IIoT system acting like a cloud AIoT. The proposed system architecture is depicted in Figure 1. In our proposal, SDADs are responsible for determining the energy data of the household appliances as well as the temperature and humidity of the room. The MQTT protocol is applied to send these various data to the edge IIoT system. All of these data are processed by the server that is a part of the edge IIoT system, and they are stored in both the local database of the edge IIoT and the cloud database of the centralized IIoT. After that, the AI apps running in the edge devices and cloud devices (basically in the workstation device) will access these databases in order to train AI models. Finally, our system is able to retrieve the real-time energy data of the household appliances, monitor that data, and identify any abnormalities in the sensor data. Data are transferred between the centralized IIoT system and the edge IIoT system, by using the MQTTS and HTTPS protocols for secure communication, both of which are based on openssl and use a self-certification mechanism. Due to the fact that our centralized IIoT system makes use of public IP addresses, the broker service, monitoring dashboard, and cloud-AI server of the centralized IIoT system can be accessed from any location at any time, using any device that is enabled for IoT. The major parts of our suggested IIoT system are depicted in Table 1. Furthermore, the proposed IIoT infrastructure is shown in Figure 2, where the data flow between the edge IIoT system and the central IIoT system can be observed more clearly.

### 2.2. Development of Smart Data Acquisition Device

The SDAD is a lightweight, sturdy, and economical device for measuring the voltage and current of home appliances as well as the temperature and humidity of the room. In Table 2, the parts of the SDAD are disclosed, including their features and applications.

In the following sections, we will explain the approaches used to determine current and voltage, in addition to the techniques by which data are exchanged between the micro-controller and the edge IIoT system.

#### 2.2.1. Voltage Measuring and Filtering Mechanisms

Appliance input voltage is measured using the “ZMPT101B AC Single Phase voltage sensor module”, which makes use of a high-precision voltage transformer. The voltage reading fluctuates when heavy power-consuming loads, such as air conditioners, heaters, and ovens are connected to the SDADs. Therefore, calibration of the ZMPT101B voltage sensor module is mandatory. However, a proposal can be found in [34] in which the execution time is longer. For convenience, in Algorithm 1, a reliable and significant voltage measuring technique is introduced.
**Algorithm 1:** Voltage measurement with noise reduction. 
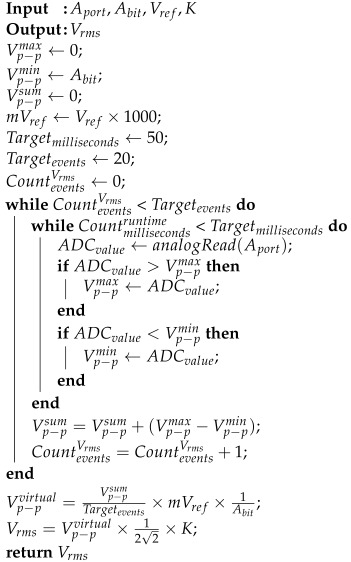


Since the supply voltage frequency is 60 Hz, at least one $V_{p-p}^{max}$ and one $V_{p-p}^{min}$ can be calculated within the time limit of $Target_{milliseconds}=50$. We specified $Target_{events}=20$ to keep the interval of measuring $V_{rms}$ at 1 s. Furthermore, a noble method for calculating the value of *K* (multiplication factor) is introduced in Equation (Equation 1). The constant *K* is significant in adjusting voltage readings to meet the proper expectation of measuring terminal voltage.
(1)$$K=\frac{V_{measured}}{V_{rms}^{initial}}$$
where $V_{measured}$ is the measured voltage from a multi-meter, $V_{rms}^{initial}$ is the RMS value of the calculated voltage by Algorithm 1 when K=1.

#### 2.2.2. Current Measuring and Filtering Mechanisms

The current of home appliances is measured by the “Gravity: Analog AC Current Sensor 20A”. This sensor is constructed by the DFROBOT based on hall current sensing principle. In the same way, in Algorithm 2, the current of home appliances is determined. According to [35], $K_{taction}=20$ was chosen as the multiplication factor of the “non-invasive 20A AC current sensor (model:SEN0211)”. We set $Target_{milliseconds}=250$ in order to maintain the interval of measuring $I_{rms}$ at 250 ms.
**Algorithm 2:** Current measurement with noise reduction. 
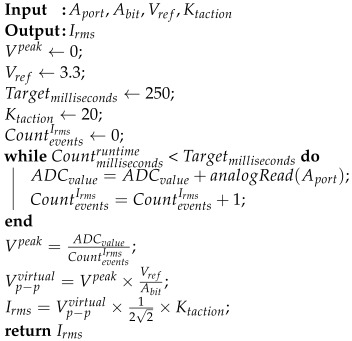


#### 2.2.3. IoT-Enabled Micro-Controllers for Energy Data Collection and Sharing

Our sensor nodes are made with ESP32-S, which are inexpensive micro-controllers with a dual-core processor and Wi-Fi and Bluetooth connectivity. Numerous peripherals are supported. These include capacitive touch, ADC, DAC, I2C, SPI, UART, I2S, PWM, and many more. They are a great choice for anyone who wants to take control of their own IoT and smart-home projects. The micro-controller’s internal registers are 32 bits in width, and its analog-to-digital converter (ADC) has a 12-bit resolution $A_{bit}=4096$. The reference DC voltage is 3.3 V. We deploy Algorithms 1 and 2 in ESP32-S. Subsequently, a local MQTT connection is established between the ESP32-S and the edge IIoT system to transmit data of date-time, voltage, and current as publish-topic in java-script format.

### 2.3. Development of Edge IIoT System

An integral aspect of the data transmission and processing infrastructure is the edge IoT system. Our proposed edge IIoT architecture includes two primary subsystems: (a) the local router and (b) the AIoT infrastructure. Conversely, the AIoT system allows for the learning and execution of AI applications everywhere, not only in the cloud. Any local router can be used to establish wireless communication with the SDADs. The local MQTT broker is developed by utilizing the Mosquitto Broker in the edge IIoT system, which is connected to the local router through an Ethernet cable. Therefore, SDAD can publish their data directly to the edge IIoT system through home/local router. In addition, local servers are constructed on edge IIoT devices, such the Jetson-nano and Raspberry Pi using the open-source Node-RED software. Other edge IIoT devices, such as computers and servers, can also be used. The data stored in the SDAD can be accessed by this local server, which can then process the data before storing them in the SQL database. In the case of edge-AIoT, data are persistently stored in the edge database. AI models are also trained after accessing these database in the edge device. Furthermore, trained AI models are deployed with the edge IIoT system. In summary, the whole system is called edge-AIoT because it gathers data from sensor nodes, processes data, extracts feature, trains the AI model and runs the AI model. On the other hand, while using cloud-AIoT, data are sent to the centralized server using MQTTS as the underlying communication protocol. This transfer occurs through worldwide internet access. The data are kept in the SQL database of the centralized server, which is often referred to as a cloud server. Additionally, artificial intelligence models are trained on the centralized server. Following this step, trained AI models are installed in the off-site edge IIoT System. Because AI models are trained and performed by the centralized IoT infrastructure, we refer to this as cloud-AIoT. The configuration setup of an edge IIoT server is illustrated in Figure 3. Data are collected from the SDADs through local MQTT as well as published to the centralized IIoT Server through a global MQTTS connection. On the other hand, in the case of the cloud-AIoT system, processing and inserting data into local databases and AI features would be removed because it would be performed by the centralized IIoT server.

### 2.4. Development of Centralized IIoT System

The workplace serves as the primary location for the majority of centralized IIoT systems. In our particular instance, we implemented an IIoT system on a workstation. We set up a public IP address on our workstation computer and install Mosquitto Broker, an open-source solution, to make a global broker system. The MQTTS protocol allows all of the edge IIoT systems to communicate with the centralized IIoT system through the global internet access. Additionally, big data, or data stored in a SQL database, are used to train AI models, and various data types are extracted from this centralized database. Once AI models have been trained, they may be used in the IIoT system remotely. Finally, a global dashboard is built to track IIoT data in real time. The configuration layout of the centralized IIoT server is depicted in Figure 4, in which each tab depicts a different home.

### 2.5. Securing Procedures of IIoT System

When it comes to our planned system for the IIoT, security is a major concern. Because we utilized public IP addresses, our network is now more exposed to potential threats. To tackle this issue, we used openssl (an open-source software) to create our own certificate authority (CA), server keys, and certificates for enabling TLS. TLS is based on Secure Socket Layer (SSL) and was developed as a replacement in response to known vulnerabilities in SSLv3. SSL is a frequently used word that, nowadays, often refers to TLS. SSL/TLS offers encryption, integrity, and authentication for data. In the sections that follow, we will talk about a complete method for keeping an IIoT system safe.

#### 2.5.1. Securing Procedures of Broker

Mosquitto provides SSL support for encrypted network connections and authentication. The CA, server, and client certificates should all have unique subject parameters; otherwise, the broker/client will not be able to distinguish between them, and the system will experience difficult-to-diagnose errors. Firstly, we need to generate a CA key and certificate to prove that we are a legitimate certificate authority. The next step is to generate a server key. A certificate sign-in request (CSR) inquiry is then created. A CA-signed server certificate can be generated from this query. In the same way, we generate a client key and a CA-signed client certificate. The procedure for securing our broker system is depicted in Figure 5. In this technique, a CA is generated using openssl’s “-x509” command. We use the “genrsa” (generating RSA) command to create an RSA (which comes from the surnames of Ron Rivest, Adi Shamir, and Leonard Adleman) key. RSA is a public-key cryptosystem to send data securely over the internet. We create a configuration file for the MQTT broker with a TLS-enabled listening port, an authorized login with username and hash-coded password information, and the deployment of “Broker-CA.crt” as the root CA certificate, “Broker-Server.key” as the server key, and “Broker-Server.crt” as the server certificate. This configuration file sets up a TLS-enabled broker on the server. “Broker-CA.crt” is the root CA certificate, “Broker-Client.key” is the client key, and “Broker-Client.crt” is the client certificate on client computers or devices, in order to establish an MQTTS connection with the broker.

#### 2.5.2. Securing Procedures of IIoT Servers

The IIoT servers were constructed in edge and centralized (also considered as cloud) devices using Node-RED. In contrast, the Node-RED editor is not secured by default. To address this, in the following Figure 6, a noble approach to secure the Node-RED server is shown. Three actions were taken to protect these servers. Firstly, HTTPS access was set up on a Node-RED server by setting up a static object settings file with a server key (Node-Sever.key) and certificate (Node-Sever.crt). As the proposed system is relying on a self-certification strategy, the CA certificate issued by the system would not be installed automatically on any of our IIoT devices, resulting in invalid CA. As a solution, the CA (Node-CA.crt) certificate is manually installed as the root CA certificate to validate our CA. We tested this method on a wide range of devices running Windows, Linux, and Android operating systems. Next, the Server Editor and Admin API are secured with an authentication method that relies on login and encrypted password credentials. Finally, the Node-RED dashboards are developed utilizing TLS and authentication mechanism.

#### 2.5.3. Securing Process of Database

Our database is designed to be automatically accessible by other programs running on the same PC. However, a global IP configuration must be specified on the computer where the database management system is installed in order for the client on other computers to connect to the database. PostgreSQL is a trustworthy and open-source database management system that we used. We implemented all three of these precautions to protect the confidentiality of our database. The primary method is to add IP addresses of the clients in the configuration file of PostgreSQL. Defining a particular port is the second. The third is to restrict distant users’ access by requiring them to utilize a certain database, along with the username and password for that database.

### 2.6. Implementation Details of AI Models

The IoT and AI are no longer optional features in the advanced IIoT system. Our research article addressed this concern by incorporating both edge and cloud AIoT solutions. In the following sections, we describe how our AI models were trained and how they were deployed in our system.

#### 2.6.1. Heterogeneous Data Extraction and Training Individual AI Models

Accessing edge and cloud databases is the first step in AI training the model. In a local environment, the database is installed on the edge IIoT system, whereas in a cloud architecture, it is installed on the centralized IIoT system. These databases can be accessed by applying “SQLAlchemy”, a python SQL toolkit. In the later phases, the system will access the big data repository and choose only the tables containing the required data. SQL queries were deployed to retrieve the necessary data by selecting the relevant tables. The tabular data are then turned into a “pandas dataframe”, a tabular data structure. Nonetheless, heterogeneous data might cause complexity and prevent the training of an effective ML/AI model due to the diversity of the data. Individual machine learning models were trained in our proposal to detect anomalies in the sensor nodes of various home appliances as well as room temperature and humidity. The energy data are also different for various household appliances. To solve this issue, a flow-chart in Figure 7 is represented that can extract heterogeneous data by separating different types of data into small groups of identical features, training individual models appropriately, and then saving these trained models with specific tag levels so that they can be used later when detecting anomalies in the sensor nodes. To detect outliers in the sensor nodes, ML models were trained independently based on the isolation forest algorithm. The assigned hyper-parameters are listed in Table 3. The outcomes of the suggested system were found to be significantly enhanced by training ML models for each data type as opposed to constructing a unified ML model using heterogeneous data.

#### 2.6.2. Performing Independent AI Models and Instantaneous Outlier Detection

In the beginning, the proposed framework establishes a connection with the edge or cloud database. Through the SQL query, the last data of room temperature, humidity and appliances’ consumed energy of a specified house are accessed as $df_{HeteroData}^{HouseName}$. The data are then scaled in $df_{HeteroData}^{Scaled}$ and the types of data in that house are counted in count. The data are segmented into $df_{Count}^{DataType}$ based on the count number from the $df{\left[count\right]}_{HeteroData}^{Scaled}$ data frame. AI models are saved by similar index numbers after training. As a result, the same count number can be assigned in $iFM_{Count}^{DataType}$ to obtain individual trained models for detecting anomalies in $dfA_{Count}^{DataType}$. After that, anomaly data are saved in $df{\left[count\right]}_{AnomalyData}^{HouseName}$ data frame. Finally, the system puts overall anomaly data into $df_{AnomalyData}^{HouseName}$, which is then saved in an edge/cloud database for future reference. Algorithm 3 is developed to instantly detect anomalies in sensor data. Database accessing, processing heterogeneous data into discrete parts, analyzing anomalies using individual AI models, and inserting outliers into the database are included in this algorithm.
**Algorithm 3:** Real-time anomaly detection by utilizing individual AI models. 
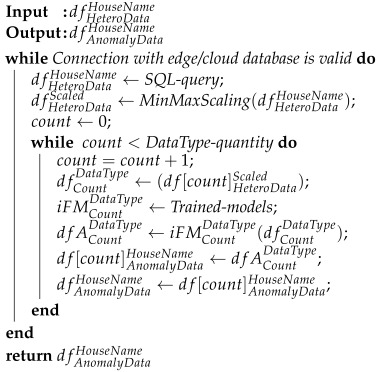


## 3. Experimental Setup

The implementation of the proposed system can be classified into three distinct categories, each of which will be described in meticulous detail below.

### 3.1. Integration of Smart Data Acquisition Device

In order to meet the needs of our specific application, we developed SDADs that are not only portable but also secure, long-lasting, and efficient. A room heater wired to an SDAD can be seen in Figure 8. All parts of the SDAD were glued together within the plastic case for maximum safety. The box was sealed properly after inserting the ESP32-S, the current sensor, and the voltage sensor. Therefore, SDAD becomes a completely safe device since none of its parts are at risk of being seen or touched.

### 3.2. Experimental Setup of Edge IIoT Systems

Our suggested solution concentrates on the edge IIoT system. Nonetheless, a pair of approaches to build the edge IIoT infrastructure is mentioned. The first is an edge-AIoT system, which includes a Jetson-nano (capable of training and running AI models), a router, and a touch-screen display. A local MQTT connection, edge IIoT server, edge database, and edge AI were all built into the Jetson-nano board. When an edge Device’s AI capabilities were inadequate, however, we made available a cloud-AIoT system. A Raspberry-Pi, router, and screen make up the cloud-AIoT. Whenever an AI model or database has to be trained or executed, the IIoT server of cloud-AIoT will make a request to the centralized IIoT System. When comparing the SDADs, edge IIoT system, and the centralized server, or so-called centralized IIoT system, the common denominator is the local/home router. As a low-cost alternative, we recommend using cloud-based AIoT systems. However, cloud-AIoT will be worthless if the internet connection is lost, and it will also outperform if the connection quality is inadequate. An effective edge-AIoT solution would be a great means of resolving such problems. In Figure 9, edge-AIoT and cloud-AIoT setups are shown.

### 3.3. Experimental Setup of Centralized IIoT System

The core of the proposed design is the centralized IIoT system. This configuration serves as a cloud-AIoT for edge IIoT systems which cannot perform AI and as a fog-computing element for edge IIoT systems which are capable of implementing AI. Figure 10 depicts the entire setup of a centralized IIoT system. Centralized IIoT becomes an integral aspect of several processes, such as gathering data and processing, artificial intelligence applications, a live streaming server of sharing AI results, performing a global broker system, and an administrative dashboard for monitoring the overall system.

## 4. Feasibility and Performance Evaluation

The aforementioned framework was evaluated according to the following three criteria: (a) privacy and security; (b) performance of AI; and (c) real-time supervision.

### 4.1. Security Verification of IIoT Infrastructure

For security purposes, only authorized users will be able to view the monitoring dashboards, as shown in Figure 11. It is also visible that the server we are accessing is HTTPS enabled. Due to the fact that a self-certified mechanism was utilized to activate TLS on the IIoT server, the server’s CA certificate had to be deployed as the root CA certificate in each IIoT devices. In Figure 12, the server certificate appears to be legitimate, which is highlighted with yellow color. Furthermore, validation of the secure connection between Node-RED server and the Mosquitto MQTT broker is shown in Figure 13 specifically marked with yellow color. The HTTPS settings of the server are refreshed every 1 h.

The global databases in the centralized IIoT system are only accessible to the authorized users distinguishing by the specified domain IP address, which greatly improves the security of these databases. As shown in Figure 14, only one device is authorized to access a particular global database. However, the IP address of that device is hidden with ash color for lab policy.

### 4.2. Performance Evaluation of AI Models

In the planned system, each residence will contain a variety of household appliances. Consequently, the databases of these homes contain different types of energy data. Additionally, the temperature and humidity of the room are recorded. In our approach, each distinct dataset in a home is applied to train AI models. In “Home#01”, for instance, there are four home appliances: a water dispenser, a refrigerator, an air conditioner, and a room heater. Individual AI models are trained for one week using temperature, humidity, and various energy consumption data. Each model was thereafter evaluated for the following several days. The red dots in Figure 15 represent anomalies in the heterogeneous data. Individual isolation forest models detect these irregularities. For closer study, a smaller portion of the plot was expanded on the right. Every second, these data were stored in the database, and their quantity climbed to 604,800. Table 4 reveals the amount and percentage of anomalies discovered during training.

Following the determination of the typical allowable range of particular data, the actual outliers of these distinct data are identified, as displayed in Table 5. In addition, the histogram displayed in Figure 16 shows how various data variables are distributed by aggregating the total number of observations into predetermined categories.

Based on Algorithm 3 the real-time anomaly detection approach is deployed to an edge IIoT system to identify outliers in the streaming heterogeneous data. This procedure is carried out for two days, and the database is dynamically updated with these abnormal data. This database of real-time anomalies is then retrieved and compared to actual anomaly data. In Figure 17, the real-time detected and realistic irregularities in the various data statistics of a house are shown. The blue “×” symbols reflect the number of anomalies discovered by the real-time anomaly detection algorithm, whereas the red “.” marks indicate the actual data of abnormalities. For clarification, a comparison was made based on the quantity of detected anomalies for both cases in Figure 18. According to this graph, the accuracy of detecting anomalies in a typical data pattern (e.g., energy consumption data of refrigerator) is greater than that in an erratic one (e.g., energy consumption data of water dispenser).

### 4.3. Real-Time Data Monitoring and Anomalies Detection

Several dashboards were established in the IIoT system in order to visualize the energy data, including energy pattern outliers across multiple appliances. In our system, dashboards can exist either at the edge or in the cloud. Evident from the dashboards, the system incorporates AI-based models for anomaly identification. Each residence in the edge IIoT system has its own dashboard, as depicted in Figure 19, which functions similarly. Timestamps are displayed as text, current measurements are displayed in a gauge (green color represents normal data, while brown color denotes anomalous data), energy data are plotted on a chart, and anomaly data are also displayed graphically. If there are discrepancies on the graph, the anomaly value will be one; otherwise, it will be zero. In addition, the administration panel is illustrated in Figure 20, where the supervisor has complete authority over all data pertaining to the residence and its appliances. The house number can be found on the left of the admin dashboard. Any of these houses can be selected, and energy as well as anomaly statistics can be seen, which are synced every second. This solution addresses the most complicated issues, such as securing and integrating IIoT systems, rapidly obtaining data from databases, extracting heterogeneous data, performing individual AI models and instantaneously spotting anomalies. This distinguishes our work from that of others.

## 5. Conclusions and Future Work

This article proposes a secure and integrated global IIoT infrastructure that comprises edge and cloud AI. The aforementioned infrastructure was developed to aggregate, analyze, and inspect heterogeneous data in real-time for the purposes of monitoring and anomaly detection. Customized SDADs are developed and implemented to collect various types of data from different sensors. To eliminate transients and distortion of the energy data, two effective algorithms are proposed. TLS protocol, hash-coded authorization, and a public IP address are used to create a globally secure broker system for the IIoT. In order to manage disparate types of large data, the PostgreSQL database system is deployed. Edge IIoT servers with HTTPS support are created so that data can be transmitted securely from sensor nodes to edge and cloud databases via the MQTTS protocol. After data of varying types are extracted from a SQL database, individual AI models can be trained. Our advanced algorithm is used in edge IIoT systems to facilitate real-time anomaly detection. Finally, a comprehensive solution for a trustworthy global AI-enabled IIoT infrastructure is completed with real-time supervisory dashboards.

Like plug-in devices, our developed SDADs are very convenient, portable devices that can be simply installed in homes. While ensuring that the specifications of the sensors are chosen to ensure smooth functioning, all parts are coated with electric-insulated glue, and a compact box contains all of them. The security of our proposed system is verified in terms of data exchange, universal dashboards, and global databases. Heterogeneous data are extracted, and multiple AI models are trained individually. The performance of real-time anomaly detection is satisfactory. From our experiments, these models have an average accuracy of around 92%. The monitoring dashboards are implemented for the individual houses and the central server, where statistics on energy and outliers are spontaneously updated every second.

In a nutshell, our proposed infrastructure is a globally accessible, capable of processing heterogeneous data, integrated with AI, secured and interconnected IIoT system for various data acquisition, outliers detection, and real-time observation, which makes it feasible and advantageous for future IIoT applications. For instance, if data discrepancies are eliminated, it is possible to accurately predict energy consumption and examine the characteristics of power utilization. In addition, if a significant number of anomalies are identified, the relay switches can be used inside SDADs to disconnect the power connection. As a conclusion, our developed system might be a practical and comprehensive solution for smart energy management systems, such as the smart grid (SG), virtual power plant (VPP), and building energy management system (BEMS). Our next approach will be to develop a BEMS based on the suggested architecture, including the use of renewable energy.

## Figures and Tables

**Figure 1 sensors-22-08980-f001:**
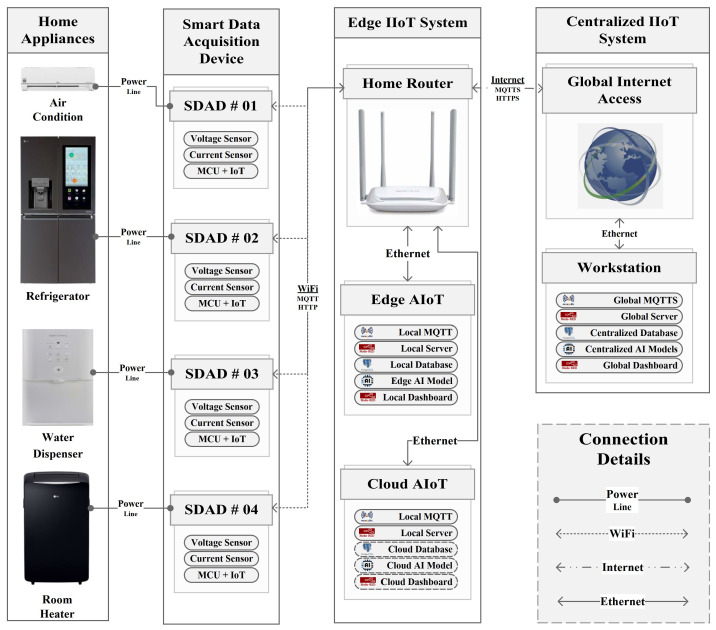
Proposed system architecture.

**Figure 2 sensors-22-08980-f002:**
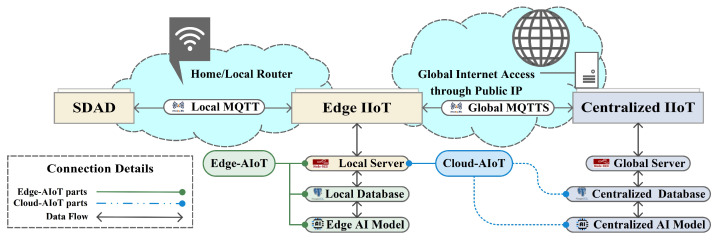
Proposed IIoT infrastructure.

**Figure 3 sensors-22-08980-f003:**
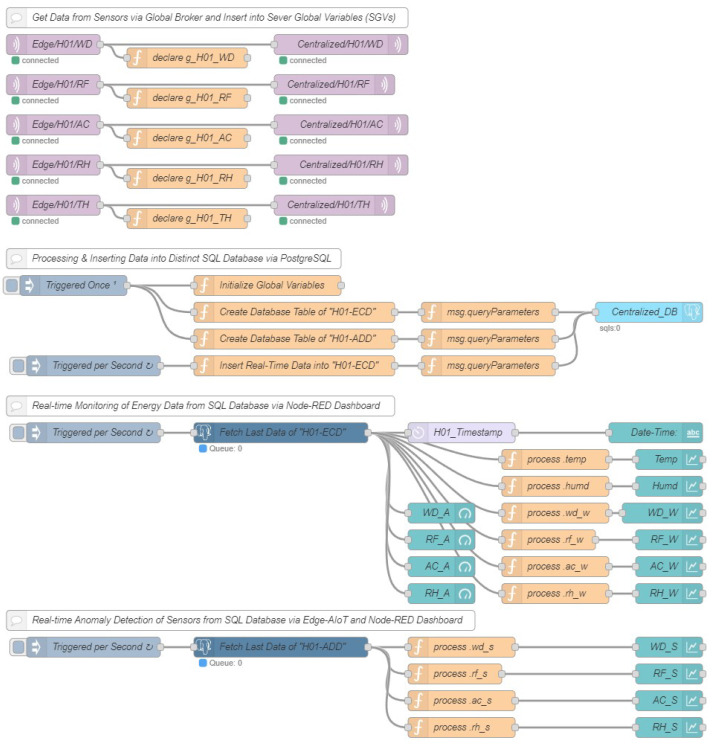
Development of an edge IIoT sever.

**Figure 4 sensors-22-08980-f004:**
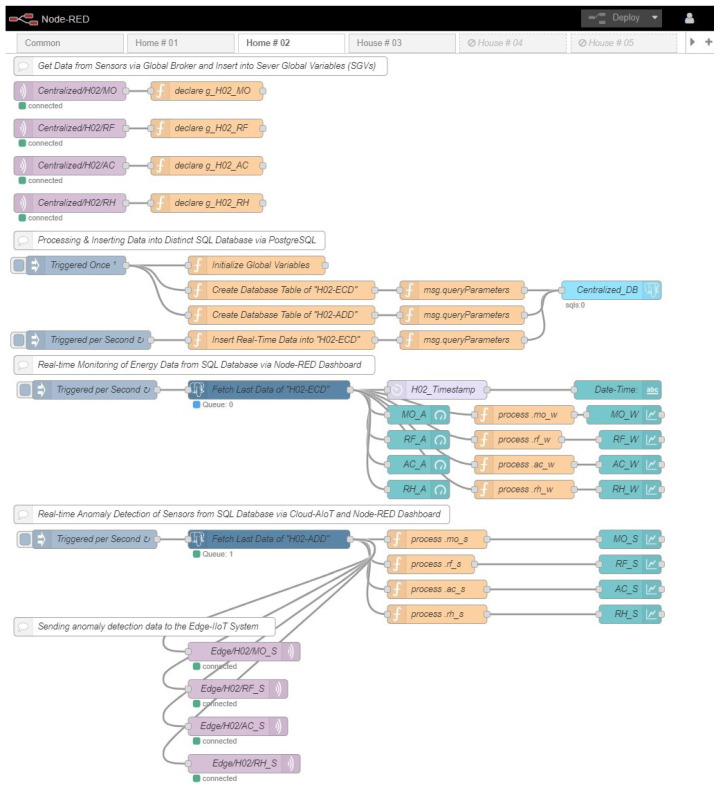
Development of the centralized IIoT server.

**Figure 5 sensors-22-08980-f005:**
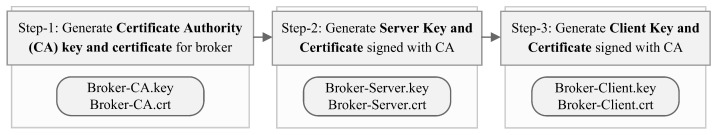
Securing steps for MQTT broker.

**Figure 6 sensors-22-08980-f006:**
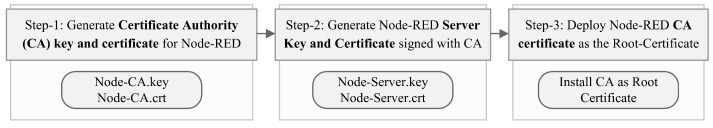
Securing steps for IIoT server.

**Figure 7 sensors-22-08980-f007:**
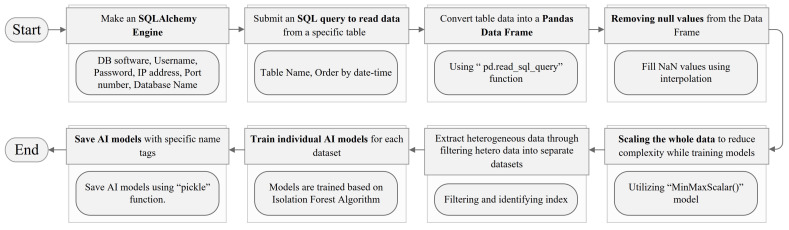
Flow-chart of training individual AI models from heterogeneous database.

**Figure 8 sensors-22-08980-f008:**
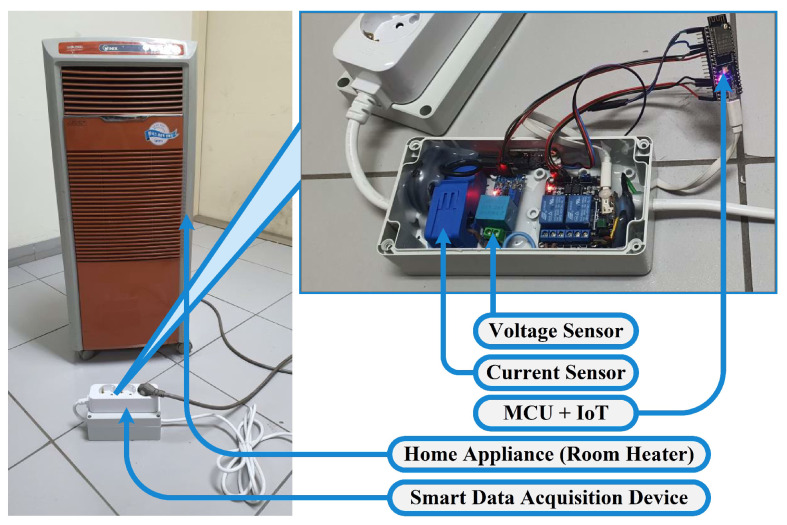
SDAD connection with a home appliance.

**Figure 9 sensors-22-08980-f009:**
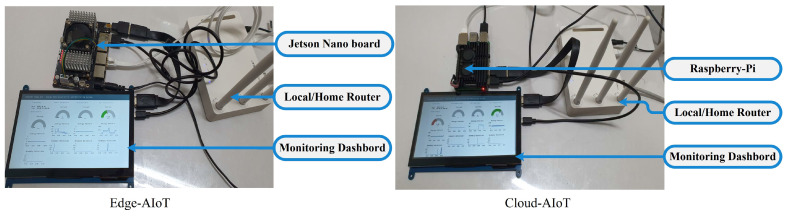
Edge and cloud AIoT setup in the edge IIoT system.

**Figure 10 sensors-22-08980-f010:**
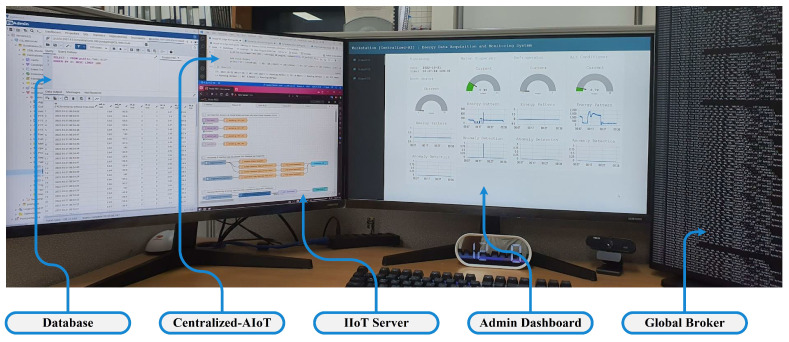
Setup of the centralized IIoT system.

**Figure 11 sensors-22-08980-f011:**
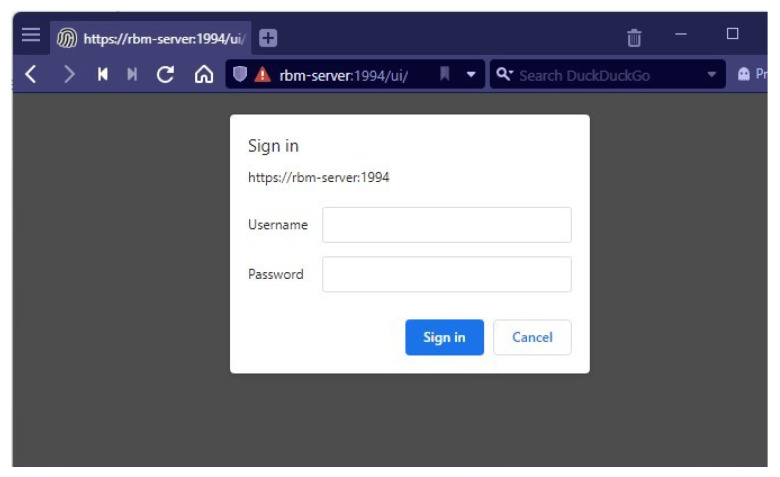
Authorized login in the dashboard.

**Figure 12 sensors-22-08980-f012:**
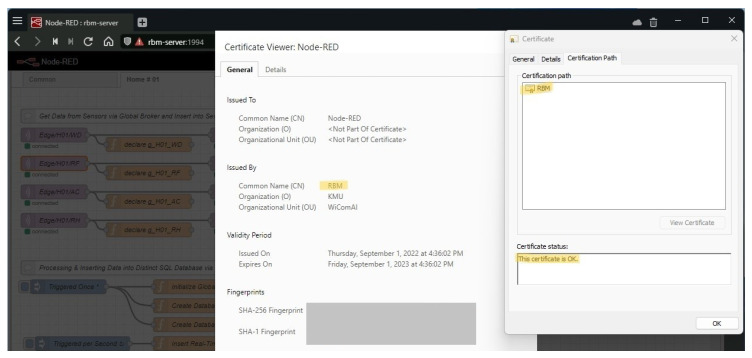
Verifying server certificate.

**Figure 13 sensors-22-08980-f013:**
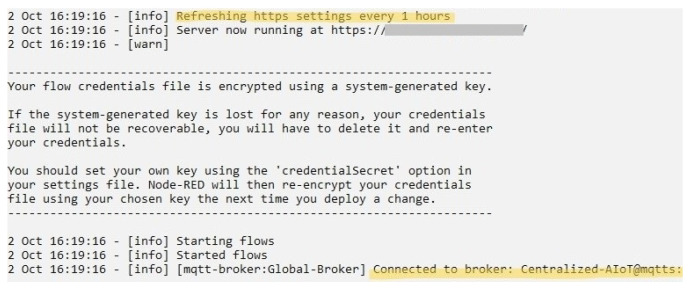
Establishing MQTT secure connection with Node-RED server.

**Figure 14 sensors-22-08980-f014:**
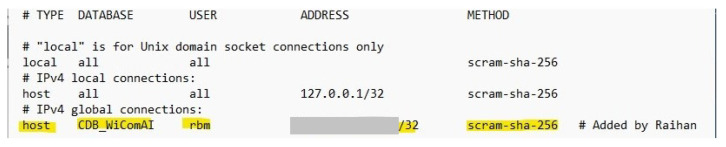
Restricting unauthorized access to the global database.

**Figure 15 sensors-22-08980-f015:**
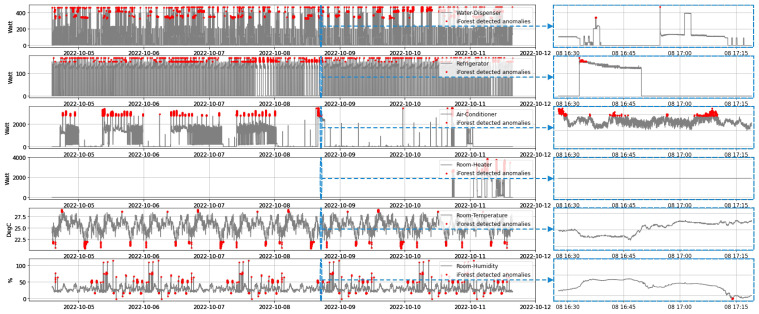
Anomalies found in the heterogeneous data of a house during training phase.

**Figure 16 sensors-22-08980-f016:**
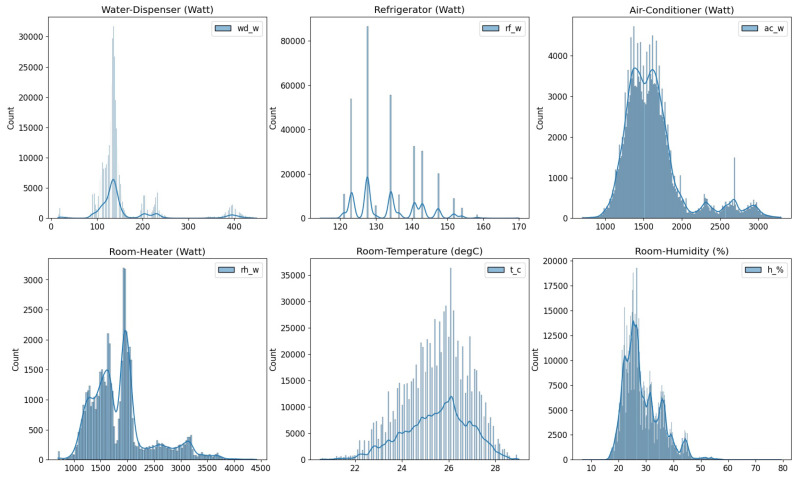
Analysis feasible range of heterogeneous data sets.

**Figure 17 sensors-22-08980-f017:**
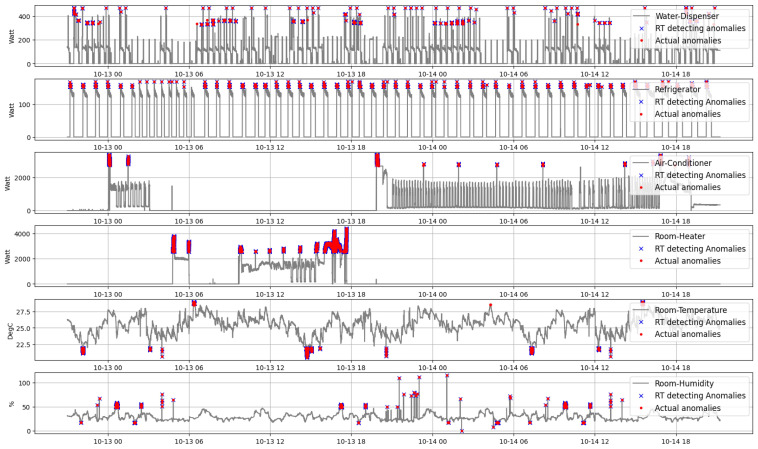
Anomalies found in the heterogeneous data of a house during testing phase.

**Figure 18 sensors-22-08980-f018:**
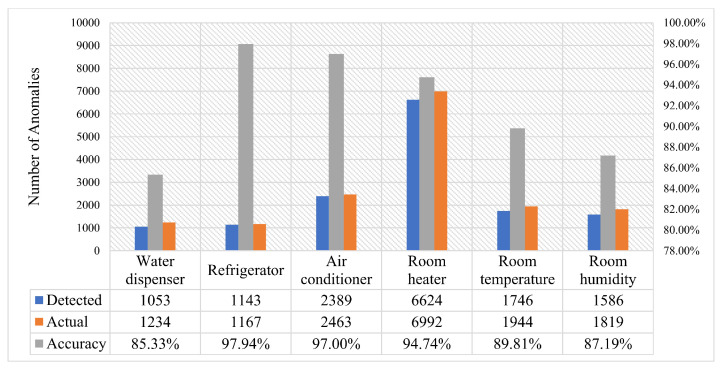
Performance analysis of the trained AI models.

**Figure 19 sensors-22-08980-f019:**
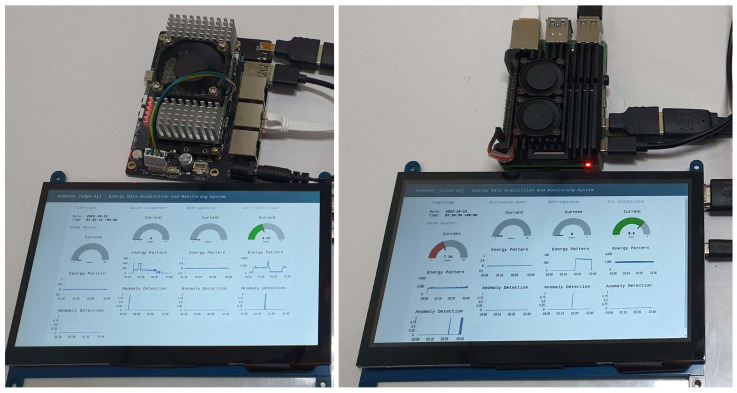
Monitoring individual house data in the edge dashboards.

**Figure 20 sensors-22-08980-f020:**
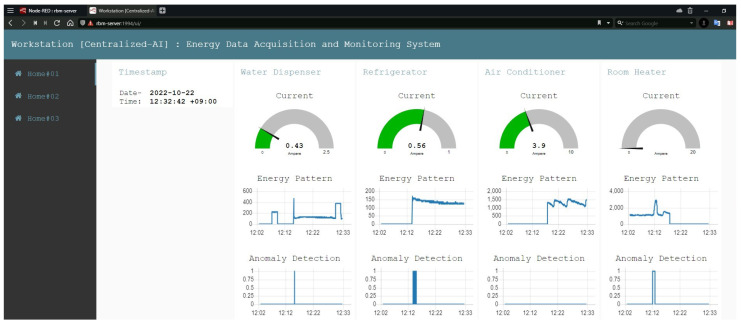
Monitoring overall system in the admin dashboard.

**Table 1 sensors-22-08980-t001:** Major parts of the proposed IIoT infrastructure.

IIoT System Major Parts	Description	Application
MQTT-Broker (Eclipse Mosquitto)	Eclipse Mosquitto is an open source message broker that implements the MQTT protocol Mosquitto is lightweight and works on single-board computers to servers Node-RED is a flow-based development tool for visual programming developed originally by IBM Node-RED connects hardware, APIs, and web services in novel ways PostgreSQL is an open source object-relational database system Gained a strong reputation for dependability, feature robustness, and efficiency NVIDIA’s Jetson-nano can run powerful AI applications Raspberry-Pi as edge server for implementing cloud AIoT system Workstation-PC are used for building centralized IIoT	Offers a simple approach to sending and receiving messages in a publish/subscribe basis
Integrating Server (Node-RED)		SDAD, edge IIoT, centralized IIoT, and SQL-database data exchange. monitoring dashboards
Database (PostgreSQL)		Storing big-data, extracting heterogeneous data
AI (Jetson-nano, Raspberry-Pi, Workstation-PC)		Train and execute AI models in the IIoT system

**Table 2 sensors-22-08980-t002:** Parts of the SDAD.

Item	Features	Application
Temperature and Humidity Sensor (AM2302)	0–100% humidity readings with 2–5% accuracy −40 to 80 °C temperature readings ±0.5 °C accuracy Measure up to 250 volts AC High precision on-board op-amp circuit Measure up to 20 amperes AC Developed based on the Hall current sensing principle Multiple analog channels MQTT protocol can be applied	Measuring room temperature and humidity
Voltage-Sensor (ZMPT101B)		Measuring terminal voltage of the Home Appliances
Current-Sensor (SEN0211)		Measuring current passing through the home-appliance
IoT enabled Micro-controller (ESP32-S)		To exchange data with edge IIoT system

**Table 3 sensors-22-08980-t003:** Parameters assigned in training isolation forest models.

Hyper-Parameters	Assigned Values	Objectives
n_estimators	100	The number of base estimators in the ensemble
contamination	0.01	The amount of contamination to define the threshold on the scores of the sample

**Table 4 sensors-22-08980-t004:** Outlier detection observation during the learning phase of isolation forest models.

Isolation Forest Models	Anomalies Detected	Percentage of Anomalies
IF model for Water Dispenser	7301	1.21%
IF model for Refrigerator	1013	0.17%
IF model for Air Conditioner	7395	1.22%
IF model for Room Heater	3977	0.66%
IF model for Room Temperature	5776	0.96%
IF model for Room Humidity	6217	1.03%

**Table 5 sensors-22-08980-t005:** Defining outlier conditions and acceptable ranges for various types of data.

Data Type	Description	Normal Range	Anomaly Condition
Water Dispenser energy data	Model: CHPI-6500L Manufacturer: Coway Model: RT17FARAEWW Manufacturer: Samsung Model: PNW1102T9FR Manufacturer: Samsung Model: PNW1102T9FR Manufacturer: Samsung House#01 Drawing Room House#01 Drawing Room	150 W to 450 W	above 450 W
Refrigerator energy data		100 W to 150 W	above 150 W
Air Conditioner energy data		730 W to 3000 W	above 3000 W
Room Heater energy data		750 W to 3450 W	above 3450 W
Temperature data		22 °C to 27.5 °C *	below 22 °C or above 27.5 °C
Humidity data		25% to 50% *	below 25% or above 50%

* Based on the accumulated data and living comfort, these ranges are considered.

## Data Availability

The public dashboard for monitoring is established. Dashboard accessible link (accessed on 1 October 2022): https://210.123.42.191:1994/ui. Due to the self-certification strategy, our CA will not be automatically distributed from the global DNS. To overcome this, click on advance and proceed. Consider “guest” to be the username and “rbm” to be the password to visit our monitoring dashboard.

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
