# Peer review of "Real-Time Energy Data Acquisition, Anomaly Detection, and Monitoring System: Implementation of a Secured, Robust, and Integrated Global IIoT Infrastructure with Edge and Cloud AI"

_sensors, 2022, doi:10.3390/s22228980_

Round 1

Reviewer 1 Report

The work is well presented and structured.

I have a minor suggestion only to include necessary codes/mathematical equations in Sequence so as the reader/research can reproduce it in future and so to improvise upon it if required. 

1. What is the main question addressed by the research? 

AI-integrated, secured IIoT infrastructure incorporating heterogeneous data collection and storing capability, global inter-communication, and a real-time anomaly detection model & study covers hardware design; development of open-source IIoT servers and databases; implementation of an interconnected global networking system; deployment of edge and cloud artificial intelligence; and development of real-time monitoring dashboards   2. Do you consider the topic original or relevant in the field? Does it address a specific gap in the field?  Original but too broad but authors have covered the implementation part very well.   3. What does it add to the subject area compared with other published material? Yes, with real-time implementation with hardware set-up.    4. What specific improvements should the authors consider regarding the methodology? What further controls should be considered? As mentioned, detailed code/mathematical steps can be provided for reproducibility

Author Response

Response to Reviewer 1 Comments

“We appreciate your thoughtful comments on our paper. Based on your valuable comments and suggestions, we have modified and updated the manuscript. We believe that the considerable revisions applied to the manuscript have made it eligible for publication in this journal. In the revised manuscript, all modifications are highlighted in yellow.”

Point 1: The work is well presented and structured. I have a minor suggestion only to include necessary codes/mathematical equations in Sequence so as the reader/research can reproduce it in future and so to improvise upon it if required.

Response 1: We revised our manuscript according to your suggestions and modified our workflow as below. We only include the essential codes (in pseudo code format), equations, and flowcharts that are relevant to our work, and we make sure to keep the bellow sequence so that it may be used by others in the future. The modified/added points are indicated in yellow here:

Stepwiseworks

Pseudocode

Equation

Flow Chart

1.        Voltage Measuring and Filtering Mechanisms

Algorithm 1

1

-

2.        Current Measuring and Filtering Mechanisms

Algorithm 2

-

-

3.        Development of Edge IIoT System

-

-

Figure 3

4.        Development of Centralized IIoT System

-

-

Figure 4

5.        Securing steps for MQTT broker

-

-

Figure 5

6.        Securing steps for IIoT server.

-

-

Figure 6

7.        Heterogeneous Data Extraction and Training Individual AI Models

-

-

Figure 7

8.        Performing Independent AI Models and Instantaneous Outlier Detection

Algorithm 3

-

-

Reviewer 2 Report

Introduction section can be reduced, seems to be too lengthy

Introduction about MQTT, PostgreSQL, Cloud AI can be introduced

Outlier detection and anamolies can be explained further

Author Response

Response to Reviewer 2 Comments

“We appreciate your thoughtful comments on our paper. Based on your valuable comments and suggestions, we have modified and updated the manuscript. We believe that the considerable revisions applied to the manuscript have made it eligible for publication in this journal. In the revised manuscript, all modifications are highlighted in yellow.”

Point 1: Introduction section can be reduced, seems to be too lengthy

Response 1: Thanks for your great suggestion. We hope your suggestion will make our work more effective. Several changes are made in the introduction section, including shortening it while maintaining the same level of essential information as before. The revised “Introduction” section is given as follows:

  1. Introduction

The Industrial Internet of Things (IIoT) is a system of interconnected devices used in industrial settings to monitor and control machinery, production lines, and human labor in real time to boost efficiency. The notion of "Industry 4.0" refers to a subset of the IIoT that places an emphasis on worker protection and increased output [1]. Nowadays, IIoT infrastructure is driven by the Internet of Things (IoT), cloud and edge computing, cyber security, AI and machine learning, and digital twin [2]. In order to decrease failures and save time and investment, companies are considering AI-powered visual insights to replace manual  inspection business models. Such as in [3], a classification model between microseismic and blasts events using the convolutional neural network (CNN) was proposed to analyze the mechanical parameters contained in microseismic events for providing accurate information of rockmass. Manufacturers can use machine learning algorithms to detect problems as soon as possible [4]. On the other hand, "Industry 5.0" refers to a future workplace environment in which humans and smart robots coexist. Industry 5.0 aims to combine cognitive computing capabilities with human intellect and resourcefulness in collaborative operations as robots in the workplace become more intelligent and interconnected [5]. AI, big data, supply chain, digital transformation, machine learning, and the IoT have all been identified as some of the most popular and widely used enablers for Industry 5.0 [6]. Moreover, the IoT enabled industries have a big impact on the environment since it uses scarce resources and lots of energy during production, usage, and recycling. In response, the area of research known as the green IoT has emerged to reduce this carbon effect [7]. Besides, the "industrial revolution" is propelled by increased connectivity, openness of data, decentralized and automated decision-making, and technological support. Industries may now collect and analyze data in real time through IoT systems for monitoring, exchanging data, and evaluating the state of the environment. When it comes to the IIoT, speed and efficiency are paramount. Large-scale deployments are required for complex systems. Therefore, it is essential that sensors maintain their performance over time while keeping  costs reasonable. If the information from these sensors is utilized to make important choices,  then latency is a measure of performance. As a popular protocol for the IoT, the message queuing telemetry transport (MQTT) is highly regarded. It’s flawless because of its small code size, seamless integration, and outstanding performance [8]. In addition, an essential feature of the IIoT for cyber-physical systems is the capacity for near real-time data streaming, which is necessary for the seamless integration of the physical and digital worlds. The manufacturer may get valuable insights from the acquired data. It is also possible to utilize the data to spot subtle problems with the manufacturing facility’s infrastructure. Furthermore, the data may be used for improvement and prediction, giving the data from the IoT devices real value. Eighty-four percent of businesses surveyed for their big data and cloud strategy cited the need for a unified platform to facilitate the transfer of information to the cloud as a top priority [9]. Furthermore, the manufacturing industry must modify its practices in response to reducing manpower, economic convenience, and ecological norms. Management of production needs adaptable decision-making procedures and the ability to self-configure. Data collected in real time from the factory floor may help guide strategy. Through real-time monitoring, any advanced system in the IIoT may make choices and  delegate authority to various stakeholders in an organization so that they can act on data in real time [10].

However, there are grave concerns relating to energy savings, real-time performance,  cohabitation, compatibility, security, and privacy in the adoption of "Industry 4.0" level IIoT infrastructure [11]. In [12], the Healthcare Industrial IoT (HealthIIoT) was proposed to monitor, track, and store patients’ healthcare information for continuous care, with data watermarked before being sent to the cloud for secure, safe, and high-quality health monitoring. But they did not utilize any AI algorithms or features. According to the [9], service-oriented architecture (SOA) was introduced to handle heterogeneous data of IoT and IIoT devices. However, they were unable to provide enough details and an appropriate solution for edge IoT sensors that communicate securely with a cloud server. The article [10] proposes methods for facilitating the digital transformation of a manufacturing line and tying such methods into the concept of the digital twin. Methods for implementing online monitoring using both traditional and IIoT sensors and collecting the resulting data have been discussed. However, this article does not go into sufficient detail on the edge computing devices and the interconnection of the vast IIoT networking architecture. Identical articles like [13–19], propose three-terminal collaborative platform (TTCP), integration of AI and IIoT technologies, Transparency Relying Upon Statistical Theory (TRUST), deep learning (DL), AI-enabled Software-Defined IIoT Network (AI-SDIN), to implement "Industry 4.0" and "Industry 5.0" facilities. Nevertheless, each of these approaches has brought its own unique perspective, ignoring the global interconnected IIoT networking system. Besides, a LoRaWAN based local IIoT infrastructure was introduced in [20] while the proposed system covers the global IIoT framework. In addition, the authors implemented a state-of-the-art open-source P2P energy trading platform in [21] that makes use of IoT and blockchain technology. It’s unexpected to discover that they declared Node-RED as their MQTT broker where as Node-RED can only act as MQTT client while making connection with a MQTT broker service like mosquitto MQTT broker according to the [22,23]. Furthermore, their proposed system is neither https nor MQTTS enabled, and customers would have to pay for a limited number of infrastructure components like private blockchain service to use it. On top of that, they haven’t integrated AI into their system. Similar articles such as [24–29], introduce interesting technologies like augmented password-only authentication and key exchange (AugPAKE), Attribute Based Encryption (ABE), Oblivious Transfer (OT), generic MQTT protocol with Mosquitto broker, and so on. Each of these publications is unaware of integrated global IIoT systems and open-source such as openss [30] based remarkable encryption protocols like utilizing self-certified certificates in TLS and SSL cryptographic protocols, which provide an extremely secure and incredibly fast communication system in an integrated IIoT infrastructure. Moreover, a simulation-based smart controller device was introduced in [31] for classifying the contracted load through a data acquisition approach, whereas the proposed SDAD is integrated and implemented on a real system. The authors in [32] developed machine learning-based abnormal voltage regulation detection in PV systems where the proposed architecture is focused on anomaly  data detection in every electrical appliance. For continuous energy flow monitoring purposes [33], the offered technique develops an AI integrated real-time monitoring system through the IIoT framework.

In this article, we implemented a globally distributed, secure, resilient, and integrated  IIoT infrastructure for real-time energy data acquisition, management, monitoring, and anomaly detection. edge and cloud AI were also integrated on the basis of "Industry 4.0" and "Industry 5.0" applications. Several algorithms, flow-charts, as well as customized devices like SDAD were exposed. Multiple edge servers, a global MQTTS broker, and an integrated cloud server were developed. Open-source based software like Node-RED, mosquitto, openssl, Visual Studio Code, etc. were utilized. In summary, the primary contribution of our research comprises:

  • Design and development of smart data acquisition devices, which are used to measure the power consumption of home appliances, focused on keeping them compact, sturdy, and economical.
  • Afterwards, HTTPS-enabled edge servers utilizing Node-RED are built for acquiring data from SDADs and inserting this data into databases.
  • Implementation of a TLS-enabled global MQTTS broker leveraging open-source software "Mosquitto" for sharing information between edge servers and cloud/centralized servers.
  • Construction of SQL databases through "PostgreSQL" in order to handle heterogeneous big data.
  • Incorporating edge and cloud AI into the system to identify outliers in the sensor readings
  • Finally, individual and centralized dashboards were implemented for real-time monitoring of the system.

On the basis of the above contributions, it is clear that our suggested system is highly advantageous in the IIoT system due to its simple architecture, secured and swift connectivity, processing capabilities of heterogeneous massive data, integration with AI, and real-time monitoring dashboards (that anyone with the proper credentials can access at any time, from any location). In addition, open-source software is used in every aspect of the proposed system, resulting in cost savings. The outline of the paper looks like this: the proposed methodology is described in Section 2. Implementations of software and hardware are demonstrated in the Section 3. Section 4 induces system evolution and experimental outcomes. In the Section 5, a brief discussion and the future direction of this study are revealed.

Point 2: Introduction about MQTT, PostgreSQL, Cloud AI can be introduced

Response 2: Thank you so much for this valuable idea. A brief description is provided in Table 1 in Section 2 about MQTT, PostgreSQL, and Cloud AI.

Furthermore, an elaborate discussion on cloud AI is mentioned in “Development Edge IIoT System” under Section 2.

Point 3: Outlier detection and anomalies can be explained further

Response 3: Thank you very much for your suggestion. In Section 4, Table 5 is added to explain how our system defines "anomalies," while Figure 16 shows how the "permissible range of operation" is validated. Moreover, the comparison and the accuracy of detecting anomalies are demonstrated in Figures 17 and 18, respectively. 

Reviewer 3 Report

This manuscript proposes an AI-integrated, secured IIoT infrastructure. It is of considerable interest. However, I thought it still has some deficiencies and I recommend to a revision before acceptable publication. Detailed comments are listed below:

--Section 1: Some existing works on artificial intelligence should be discussed, then highlight the origin of the presented work, such as: Dong, L.-j.; Tang, Z.; Li, X.-b.; Chen, Y.-c.; Xue, J.-c. Discrimination of mining microseismic events and blasts using convolutional neural networks and original waveform. J. Cent. South Univ. 2020, 27, 3078-3089; Sisinni, E.; Saifullah, A.; Han, S.; Jennehag, U.; Gidlund, M. Industrial Internet of Things: Challenges, Opportunities, and Directions. IEEE Transactions on Industrial Informatics 2018, 14, 4724-4734; Hossain, M.S.; Muhammad, G. Cloud-assisted Industrial Internet of Things (IIoT) - Enabled framework for health monitoring. Computer Networks 2016, 101, 192-202.

--Section 2: The AI models are separately trained using heterogeneous data, which are not shown in the experiment, because only energy data are used to detect anomalies in Figure 15.

--Section 2: Whether the AI models can update with new data recorded by sensors or not?

--Section 4: How are the anomalies defined?

--Section 4: What does brown mean in the dashboard of house#2 (Figure 18).

--Section 5: What is the feasibility of applying this framework to family safety.

Author Response

Response to Reviewer 3 Comments

“We appreciate your thoughtful comments on our paper. Based on your valuable comments and suggestions, we have modified and updated the manuscript. We believe that the considerable revisions applied to the manuscript have made it eligible for publication in this journal. In the revised manuscript, all modifications are highlighted in yellow.”

Point 1: Some existing works on artificial intelligence should be discussed, then highlight the origin of the presented work, such as: Dong, L.-j.; Tang, Z.; Li, X.-b.; Chen, Y.-c.; Xue, J.-c. Discrimination of mining microseismic events and blasts using convolutional neural networks and original waveform. J. Cent. South Univ. 2020, 27, 3078-3089; Sisinni, E.; Saifullah, A.; Han, S.; Jennehag, U.; Gidlund, M. Industrial Internet of Things: Challenges, Opportunities, and Directions. IEEE Transactions on Industrial Informatics 2018, 14, 4724-4734; Hossain, M.S.; Muhammad, G. Cloud-assisted Industrial Internet of Things (IIoT) - Enabled framework for health monitoring. Computer Networks 2016, 101, 192-202.

Response 1: Thank you very much for your suggestion. After reviewing these research works, we cited those papers in this article. The following is an updated version of our introduction. The most recent citations are highlighted here. The modified "Introduction" is presented as below.

  1. Introduction

The Industrial Internet of Things (IIoT) is a system of interconnected devices used in industrial settings to monitor and control machinery, production lines, and human labor in real time to boost efficiency. The notion of "Industry 4.0" refers to a subset of the IIoT that places an emphasis on worker protection and increased output [1]. Nowadays, IIoT infrastructure is driven by the Internet of Things (IoT), cloud and edge computing, cyber security, AI and machine learning, and digital twin [2]. In order to decrease failures and save time and investment, companies are considering AI-powered visual insights to replace manual  inspection business models. Such as in [3], a classification model between microseismic and blasts events using the convolutional neural network (CNN) was proposed to analyze the mechanical parameters contained in microseismic events for providing accurate information of rockmass. Manufacturers can use machine learning algorithms to detect problems as soon as possible [4]. On the other hand, "Industry 5.0" refers to a future workplace environment in which humans and smart robots coexist. Industry 5.0 aims to combine cognitive computing capabilities with human intellect and resourcefulness in collaborative operations as robots in the workplace become more intelligent and interconnected [5]. AI, big data, supply chain, digital transformation, machine learning, and the IoT have all been identified as some of the most popular and widely used enablers for Industry 5.0 [6]. Moreover, the IoT enabled industries have a big impact on the environment since it uses scarce resources and lots of energy during production, usage, and recycling. In response, the area of research known as the green IoT has emerged to reduce this carbon effect [7]. Besides, the "industrial revolution" is propelled by increased connectivity, openness of data, decentralized and automated decision-making, and technological support. Industries may now collect and analyze data in real time through IoT systems for monitoring, exchanging data, and evaluating the state of the environment. When it comes to the IIoT, speed and efficiency are paramount. Large-scale deployments are required for complex systems. Therefore, it is essential that sensors maintain their performance over time while keeping  costs reasonable. If the information from these sensors is utilized to make important choices,  then latency is a measure of performance. As a popular protocol for the IoT, the message queuing telemetry transport (MQTT) is highly regarded. It’s flawless because of its small code size, seamless integration, and outstanding performance [8]. In addition, an essential feature of the IIoT for cyber-physical systems is the capacity for near real-time data streaming, which is necessary for the seamless integration of the physical and digital worlds. The manufacturer may get valuable insights from the acquired data. It is also possible to utilize the data to spot subtle problems with the manufacturing facility’s infrastructure. Furthermore, the data may be used for improvement and prediction, giving the data from the IoT devices real value. Eighty-four percent of businesses surveyed for their big data and cloud strategy cited the need for a unified platform to facilitate the transfer of information to the cloud as a top priority [9]. Furthermore, the manufacturing industry must modify its practices in response to reducing manpower, economic convenience, and ecological norms. Management of production needs adaptable decision-making procedures and the ability to self-configure. Data collected in real time from the factory floor may help guide strategy. Through real-time monitoring, any advanced system in the IIoT may make choices and  delegate authority to various stakeholders in an organization so that they can act on data in real time [10].

However, there are grave concerns relating to energy savings, real-time performance,  cohabitation, compatibility, security, and privacy in the adoption of "Industry 4.0" level IIoT infrastructure [11]. In [12], the Healthcare Industrial IoT (HealthIIoT) was proposed to monitor, track, and store patients’ healthcare information for continuous care, with data watermarked before being sent to the cloud for secure, safe, and high-quality health monitoring. But they did not utilize any AI algorithms or features. According to the [9], service-oriented architecture (SOA) was introduced to handle heterogeneous data of IoT and IIoT devices. However, they were unable to provide enough details and an appropriate solution for edge IoT sensors that communicate securely with a cloud server. The article [10] proposes methods for facilitating the digital transformation of a manufacturing line and tying such methods into the concept of the digital twin. Methods for implementing online monitoring using both traditional and IIoT sensors and collecting the resulting data have been discussed. However, this article does not go into sufficient detail on the edge computing devices and the interconnection of the vast IIoT networking architecture. Identical articles like [13–19], propose three-terminal collaborative platform (TTCP), integration of AI and IIoT technologies, Transparency Relying Upon Statistical Theory (TRUST), deep learning (DL), AI-enabled Software-Defined IIoT Network (AI-SDIN), to implement "Industry 4.0" and "Industry 5.0" facilities. Nevertheless, each of these approaches has brought its own unique perspective, ignoring the global interconnected IIoT networking system. Besides, a LoRaWAN based local IIoT infrastructure was introduced in [20] while the proposed system covers the global IIoT framework. In addition, the authors implemented a state-of-the-art open-source P2P energy trading platform in [21] that makes use of IoT and blockchain technology. It’s unexpected to discover that they declared Node-RED as their MQTT broker where as Node-RED can only act as MQTT client while making connection with a MQTT broker service like mosquitto MQTT broker according to the [22,23]. Furthermore, their proposed system is neither https nor MQTTS enabled, and customers would have to pay for a limited number of infrastructure components like private blockchain service to use it. On top of that, they haven’t integrated AI into their system. Similar articles such as [24–29], introduce interesting technologies like augmented password-only authentication and key exchange (AugPAKE), Attribute Based Encryption (ABE), Oblivious Transfer (OT), generic MQTT protocol with Mosquitto broker, and so on. Each of these publications is unaware of integrated global IIoT systems and open-source such as openss [30] based remarkable encryption protocols like utilizing self-certified certificates in TLS and SSL cryptographic protocols, which provide an extremely secure and incredibly fast communication system in an integrated IIoT infrastructure. Moreover, a simulation-based smart controller device was introduced in [31] for classifying the contracted load through a data acquisition approach, whereas the proposed SDAD is integrated and implemented on a real system. The authors in [32] developed machine learning-based abnormal voltage regulation detection in PV systems where the proposed architecture is focused on anomaly  data detection in every electrical appliance. For continuous energy flow monitoring purposes [33], the offered technique develops an AI integrated real-time monitoring system through the IIoT framework.

In this article, we implemented a globally distributed, secure, resilient, and integrated  IIoT infrastructure for real-time energy data acquisition, management, monitoring, and anomaly detection. edge and cloud AI were also integrated on the basis of "Industry 4.0" and "Industry 5.0" applications. Several algorithms, flow-charts, as well as customized devices like SDAD were exposed. Multiple edge servers, a global MQTTS broker, and an integrated cloud server were developed. Open-source based software like Node-RED, mosquitto, openssl, Visual Studio Code, etc. were utilized. In summary, the primary contribution of our research comprises:

  • Design and development of smart data acquisition devices, which are used to measure the power consumption of home appliances, focused on keeping them compact, sturdy, and economical.
  • Afterwards, HTTPS-enabled edge servers utilizing Node-RED are built for acquiring data from SDADs and inserting this data into databases.
  • Implementation of a TLS-enabled global MQTTS broker leveraging open-source software "Mosquitto" for sharing information between edge servers and cloud/centralized servers.
  • Construction of SQL databases through "PostgreSQL" in order to handle heterogeneous big data.
  • Incorporating edge and cloud AI into the system to identify outliers in the sensor readings
  • Finally, individual and centralized dashboards were implemented for real-time monitoring of the system.

On the basis of the above contributions, it is clear that our suggested system is highly advantageous in the IIoT system due to its simple architecture, secured and swift connectivity, processing capabilities of heterogeneous massive data, integration with AI, and real-time monitoring dashboards (that anyone with the proper credentials can access at any time, from any location). In addition, open-source software is used in every aspect of the proposed system, resulting in cost savings. The outline of the paper looks like this: the proposed methodology is described in Section 2. Implementations of software and hardware are demonstrated in the Section 3. Section 4 induces system evolution and experimental outcomes. In the Section 5, a brief discussion and the future direction of this study are revealed.

Point 2: The AI models are separately trained using heterogeneous data, which are not shown in the experiment, because only energy data are used to detect anomalies in Figure 15.

Response 2: Thank you for your observation. We strongly agree with your findings. Our updated system is considering three kinds of data, such as temperature, humidity, and various energy patterns. We are attaching major modifications for your consideration.

Modification-1: (Section-2) Introducing temperature and humidity sensor.

Modification-2: (Section-2) Figure 7 is updated for heterogeneous data extraction and training individual AI models.

Modification-3: (Section-2) Modifying Section-2.6.2 for performing independent AI models and instantaneous outlier detection

Modification-4: (Section-4) Figure 15 and Table 4 are updated for "Performing Independent AI Models and Instantaneous Outlier Detection."

Modification-5: (Section-4) the comparison and the accuracy of detecting anomalies are demonstrated in Figures 17 and 18, respectively, which are also modified.

Modification-6: (Section-5) The average accuracy of the trained models was recalculated and found to be 92%.

Point 3: Whether the AI models can update with new data recorded by sensors or not?

Response 3: Thank you very much for your concern. Yes, the trained AI models can update anomaly data with the new data recoded by the sensors based on Algorithm 3, which is mentioned in Section 2. For more clarification, we are sharing a screenshot of our continuously running AI program based on Algorithm 3 on the right side. You can find that it is a continuous and real-time process.

Point 4: How are the anomalies defined?

Response 4: Thank you very much for your suggestion. In Section 4, Table 5 is added to explain how our system defines "anomalies," while Figure 16 shows how the "permissible range of operation" is validated. 

Point 5: What does brown mean in the dashboard of house#2 (Figure 18).

Response 5: Thanks for your concern in this regard. We explained the meaning of green (normal data) and brown (abnormal data or data beyond the normal range) in Section 4 as below.

Point 6: What is the feasibility of applying this framework to family safety.

Response 6: Thanks for your great thoughts. Our framework should be a feasible solution to family safety because:

  • Each sensor's data is securely transferred to the database through the MQTTS protocol.
  • The database is also secured by limiting access only to the device where AI programs are running.
  • Monitoring dashboards use HTTPS protocols, including a specified username and password. Therefore, other people cannot see the data.

To clarify these issues, we have modified Section 5 (Conclusion and Future Work) as below.
